# A Coupled Variational System for Image Decomposition along with Edges Detection

**Jianlou Xu** [1,*], **Yuying Guo** [1], **Yan Hao** [1] **and Leigang Huo** [2]

1   School of Mathematics and Statistics, Henan University of Science and Technology, Luoyang 471023, China
2   School of Computer and Information Engineering, Nanning Normal University, Nanning 530001, China
*   Correspondence: xujianlou@haust.edu.cn

**Abstract:** In order to better decompose the images and protect their edges, in this paper, we proposed a coupled variational system consisting of two steps. The first step, an improved weighted variational model is introduced to obtain the cartoon and texture. Using the obtained cartoon image, in the second step, a new vector function is obtained for describing the pseudo edge of the considered image by one Tikhonov regularization variational model. Because Tikhonov regularization model is equivalent to carrying out a Gaussian linear filtering, the obtained vector function is smoother. To solve the coupled system, we give the alternating direction method, primal-dual method and Gauss-Seidel iteration. Using the coupled system, we can not only separate out the cartoon and texture parts, but also extract the edge. Extensive numerical experiments are given to show the effectiveness of the proposed method compared with other variational methods.

**Keywords:** image decomposition; image restoration; total variation; texture; cartoon

## 1. Introduction

Image decomposition is an important research topic in the field of image processing. The purpose is to decompose a given image into the different parts. For example, for a noise-free image, we can split it into cartoon and texture; for a noisy smooth image, we can split it into cartoon and noise. The cartoon part (also called structure component) is ideally piecewise smooth parts corresponding to the main objects; the texture part usually refers to the fine scale details consisting of repetitive patterns presented in the original image.

Variational methods for image decomposition have become very popular in recent years. Their aims are to find the appropriate priors to describe the cartoon and texture components so that the two components have respective small norms in their corresponding functional spaces [1–15]. For example, a classical variational model was proposed by Rudin et al. [1]. The cartoon component belongs to the bounded variation space and the noise or small scale texture component is in $L^2$. The model denoises well for the piecewise-constant images while preserving sharp edges. However, some details can be destroyed. Consequently, Meyer [2] introduced the dual spaces of bounded variation space to model the oscillating patterns and pointed out that the oscillating patterns should have small norm in these dual spaces. However, Meyer's model can not directly give its numerical solutions. To solve the difficulty, the authors in [3–6] designed different approximation models. Other methods based on total variation (TV) can be seen in [7–15]. As is well known, TV regularization easily tends to make the staircase effect. To reduce the staircase effect, some higher-order and nonlocal variational models were discussed in [16–21] for image decomposition.

Different from the above approaches, Xu et al. [22] proposed a relative TV regularization model with the windowed TV norm. Based on sparse approximation theory, Starck et al. [23] and Cai et al. [24] designed two different dictionaries and tight frames for the cartoon and texture respectively. Based on the discriminative patch recurrence prior, Xu

et al. [25] constructed an effective approach for cartoon-texture decomposition. Through the relative variation of gradient, Buades et al. [26] designed a fast filter method for cartoon-texture decomposition. By increasing the size of the parameter, multiscale methods for image decomposition were studied in [27–30]. Using the texture similarity, some low rank techniques for image decomposition were proposed in [31–35].

With the development of deep learning, some deep unfolding learning methods for image decomposition were studied in [36,37]. However, on the one hand, it is difficult to find the original cartoon and texture data for training, on the other hand, deep unfolding networks have to adopt the input and the output of each stage as an image, which inevitably results in feature-to-image information distortion [38]. So it is difficult for us to get good decomposition results based on deep learning method. In addition, [36,37] are based on $TV - L^1$ model which is not suitable for image decomposition [2].

Recently, Surya Prasath et al. [39] introduced a coupled system for image restoration by combining a constrained partial differential equation (PDE) with weighted TV. Substituting the first equation of the coupled system in [39] by the weighted $TV - L^1$ regularization model, Moreno et al. [40] proposed a new coupled system and gained some better image denoising and decomposition results. Because $TV - L^1$ model is not suitable for additive Gaussian noise, a higher order variational system was proposed by Xu et al. [11]. Inspired by [11,39,40], in this paper, we give a new different coupled system which contains two variational minimal models. One is for cartoon-texture decomposition model which adopts $H^{-1}$ norm to describe the oscillation part, the other is a Tikhonov regularization model for the smoothed vector field. These two models are intertwined, that is, the smooth vector field can affect the cartoon-texture decomposition result. In turn, the decomposed cartoon image can also affect the vector field result. Some numerical examples will be given to highlight the decomposition nature of the proposed model along with some comparison results.

In the rest of this paper, we review the prior works in Section 2. In Section 3, we will give the proposed model and the corresponding numerical algorithm. Section 4 will provide some numerical experiments and analysis. At the last, the conclusions are given.

## 2. Prior Work

To improve the denoising capability, Perona-Malik (PM) [41] discussed the following PDE of anisotropic diffusion

$$\frac{\partial u}{\partial t} = \text{div}(h(|\nabla u|)\nabla u) \tag{1}$$

where $h(|\nabla u|) = \frac{1}{1+K|\nabla u|^2}$, $K > 0$ is a contrast parameter. Due to the impact of noise, the computation of gradient is not robust to outliers. Hence, with the aid of the Gaussian convolution, Catté et al. [42] introduced the modified model

$$\frac{\partial u}{\partial t} = \text{div}(h(|G_\sigma * \nabla u|)\nabla u) \tag{2}$$

where $G_\sigma(x) = (2\pi\sigma)^{-1}\exp\left(-\left(|x|^2/2\sigma\right)\right)$ is the Gaussian kernel function and $*$ means convolution. Because Gaussian smoothing is equivalent to solving the heat diffusion equation, Surya Prasath et al. [39] used an auxiliary variable $\omega$ to split Equation (1) into the following PDEs system

$$\frac{\partial u}{\partial t} = \text{div}(h(\omega)\nabla u) \tag{3}$$

$$\frac{\partial \omega}{\partial t} = \lambda \text{div}(\nabla \omega) + (1-\lambda)(|\nabla u| - \omega) \tag{4}$$

where $\lambda$ is a balancing parameter. Later, the authors in [40] used a weighted $TV - L^1$ model to replace Equation (3) and considered the following coupled variational model

$$\min_u \int_\Omega h(\omega)|\nabla u|dx + \int_\Omega \mu(x)|u - f|dx \tag{5}$$

$$\frac{\partial \omega}{\partial t} = \lambda \mathrm{div}(\nabla \omega) + (1 - \lambda)(|\nabla u| - \omega) \tag{6}$$

where $\mu(x)$ is an adaptive parameters, $\omega$ is called the pseudo edge function.

Because Equation (5) is not suitable for Gaussian noise, and the solutions of Equations (4) and (6) are not the same as the form of Equation (2), Xu et al. [11] proposed the following high order coupled model

$$\min_{u,v} \alpha_1 \int_\Omega h(\omega)|\nabla u - v|dx + \alpha_2 \int_\Omega |\nabla v|dx + \frac{\beta}{2}\int_\Omega |u - f|^2 dx \tag{7}$$

$$\min_\omega \int_\Omega |\nabla \omega|^2 dx + \lambda \int_\Omega |\nabla u - \omega|^2 dx \tag{8}$$

where $\alpha_1$, $\alpha_2$, $\beta$, $\lambda$ are positive parameters, $f$ is the observed image.

## 3. The Proposed Model and Algorithm

### 3.1. The Proposed Model

Meyer [2] pointed out that the oscillatory components should use a weaker norm to describe, Equations (3), (5) and (7) are not suitable for image decomposition, and Meyer gave the following model

$$\min_{u,v} \int_\Omega |\nabla u|dx + \tau\|v\|_G \quad , \quad s.t. \quad f = u + v \tag{9}$$

where $\|\cdot\|_G$ represents the dual $G$ norms (see details in [2]).

Because Equation (9) is hard to compute its numerical solution, Vese and Osher [3] gave an approximation model of (9) as follows.

$$\min_{u,g} \int_\Omega |\nabla u|dx + \frac{\beta}{2}\int_\Omega \left|f - (u + \mathrm{div}g)\right|^2 dx + \frac{\mu}{2}\left[\int_\Omega \left(\sqrt{g_1^2 + g_2^2}\right)^p dx\right]^{\frac{1}{p}} \tag{10}$$

where $\beta$, $\mu$ are positive tuning parameters, $p > 0$, $g = (g_1, g_2)$. $\mathrm{div}g = \mathrm{div}(g_1, g_2)$ represents the texture component and $u$ represents the cartoon component.

With the Hodge decomposition of $g$ in [4], then $g = \nabla P + Q$, where $P$ is a single-valued function and $Q$ is a divergence-free vector field. Letting $v = \mathrm{div}g$, we have $P = \Delta^{-1}v$, $g = \nabla(\Delta^{-1}v)$ by ignoring $Q$, where $\Delta = \mathrm{div}(\nabla)$ is the Laplace operator, $\Delta^{-1}$ its inverse operator. Setting $p = 2$, then Equation (10) can be turned into the following model:

$$\min_{u,v} \int_\Omega |\nabla u|dx + \frac{\beta}{2}\int_\Omega \left|f - (u + v)\right|^2 dx + \frac{\mu}{2}\left(\int_\Omega \left|\nabla\left(\Delta^{-1}\right)v\right|^2 dx\right)^{\frac{1}{2}} \tag{11}$$

Combining Equations (11) and (8), we study the following coupled variational system for image decomposition:

$$\min_{u,v} \int_\Omega h(\omega)|\nabla u|dx + \frac{\mu}{2}\int_\Omega \left|\nabla\left(\Delta^{-1}\right)(v)\right|^2 dx + \frac{\beta}{2}\left\|f - u - v\right\|_2^2 \tag{12}$$

$$\min_\omega \lambda \int_\Omega |\nabla \omega|^2 dx + (1 - \lambda)\int_\Omega \left|\nabla u - \omega\right|^2 dx \tag{13}$$

where $h(\omega)$ is defined as Equation (1), $\mu$, $\beta$, $\lambda(0 < \lambda < 1)$ are the positive parameters, $f$ is an observed image, $u$ is the cartoon image, and $v$ is the texture image or noise. Similar to the definition of [11,39,40], the norm of $\omega$ is called the pseudo edge function.

The first term in (12) contains an edge indicator function $h(\omega)$ and when $\omega \to \infty$, $h \to 0$, it can preserve edges; when $\omega \to 0$, then $h \to 1$, it can remove some small details. The second term in (12) is the $H^{-1}$ norm for describing the texture. The third term in (12) is a residual term. Compared with [11,39,40], mathematically, the proposed model is more suitable for image decomposition.

### 3.2. Algorithm

We now firstly give the detail algorithm for (12), which can be turned into the following two coupled problems:

For $\omega, v$ fixed, we solve

$$\min_{u} \int_{\Omega} h(\omega)|\nabla u|dx + \frac{\beta}{2}\left\|f - u - v\right\|_{2}^{2} \tag{14}$$

For $u$ fixed, we have

$$\min_{v} \mu \int_{\Omega} \left|\nabla\left(\Delta^{-1}\right)(v)\right|^{2}dx + \frac{\beta}{2}\left\|f - u - v\right\|_{2}^{2} \tag{15}$$

For (14), using the Legendre-Fenchel's transformation, it becomes

$$\min_{u} \max_{\mathbf{p} \in B}\langle\nabla u, \mathbf{p}\rangle + \frac{\beta}{2}\left\|f - u - v\right\|_{2}^{2}, \tag{16}$$

where $B = \left\{\mathbf{p} = (p_1, p_2)^{T}|\ \|\mathbf{p}\|_{\infty} \leq h(\omega)\right\}$, $\|\mathbf{p}\|_{\infty} = \sup\limits_{x \in \Omega}(p_1^2 + p_2^2)^{\frac{1}{2}}$.

Then (16) can be solved by the primal-dual method

$$\begin{aligned} \mathbf{p}^{k+1} &= \text{proj}_{B}\left(\mathbf{p}^{k} + \theta\left(\nabla u^{k}\right)\right), \\ u^{k+1} &= \frac{u^{k} + \tau\beta\left(f - v^{k}\right) + \tau\text{div}\left(\mathbf{p}^{k+1}\right)}{1 + \tau\beta}, \end{aligned} \tag{17}$$

where $\theta, \tau$ are the positive parameters.

Minimizing (15) with respect to $v$ results in the following Euler-Lagrange equation:

$$(\mu - \beta\Delta)v^{k+1} = \beta\Delta\left(u^{k+1} - f\right), \tag{18}$$

which can be solved by Gauss-Seidel iteration.

For (13), the corresponding Euler-Lagrange equation is

$$(1 - \lambda - \lambda\Delta)\omega^{k+1} = (1 - \lambda)\nabla u^{k+1} \tag{19}$$

and we can solve it using the Gauss-Seidel iteration.

From the above derivation, we now summarize the detail algorithm for the coupled system as Algorithm 1:

---

**Algorithm 1** Given the initial value.

---

　　Step 1: Compute $\mathbf{p}^{k+1}, u^{k+1}$ by (17),
　　Step 2: Compute $v^{k+1}$ by (18),
　　Step 3: Compute $\omega^{k+1}$ by (19),
　Until: A stopping criterion is satisfied; otherwise set $k = k + 1$ and return to Step 1.

---

## 4. Experimental Results and Discussion

In this section, we would like to show some numerical results for different methods. The best results are shown according to the stopping condition $\max\left(\frac{\|u^{k+1}-u^k\|_2}{\max(1,\|u^k\|_2)}, \frac{\|v^{k+1}-v^k\|_2}{\max(1,\|v^k\|_2)}\right) \leq \varepsilon$ or the max iteration number.

### 4.1. The Parameters Selection Discussion

For proposed method, these parameters $K, \mu, \beta, \tau, \lambda, \theta$ need to be given to run the algorithm. For $K$, we find that when it is smaller, the cartoon image will become smoother. Once $K$ is determined, we fix $\tau = 2$ and change the other parameters, we find that when $\beta \leq 0.05$ and it is smaller and smaller, the cartoon image will become smoother and smoother. To see these experimental phenomena more clearly, we take the Finger image as an example and fix $\beta = 0.02$, $\mu = 0.1$, $\lambda = 0.2$, $\theta = 0.5$, $\tau = 2$ to change $K$ and obtain the flowing results in Figure 1.

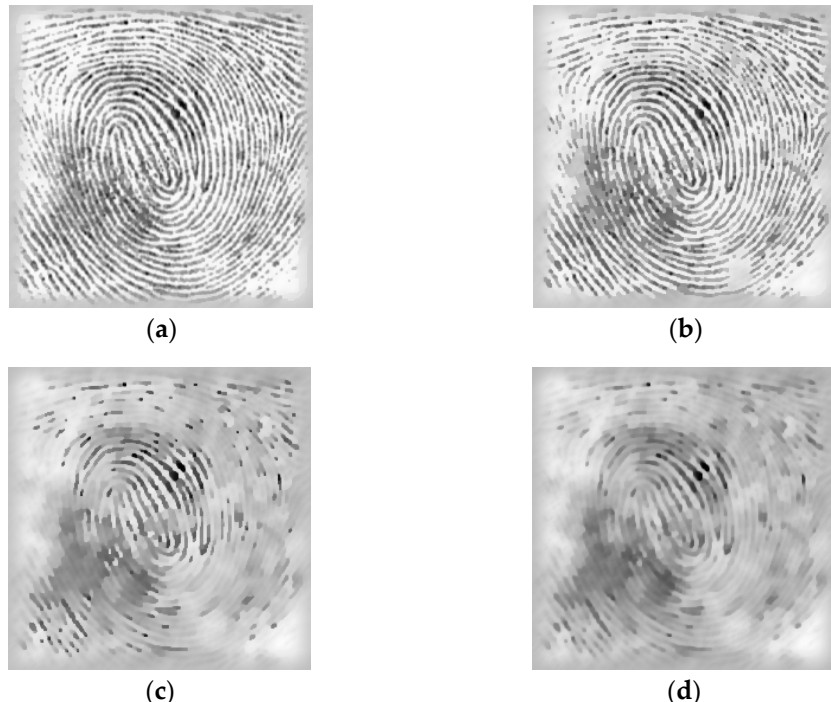

**Figure 1.** Results for different $K$. (**a**) $K = 0.1$; (**b**) $K = 0.01$; (**c**) $K = 0.001$; (**d**) $K = 0.0001$.

For $\lambda$, if it is larger and larger, the pseudo edge $|\omega|$ image will get thicker and thicker, the cartoon becomes more and more blurred. In short, we choose $K = 0.001$ in the following experiments, other parameters are adjusted according to different test images.

The test images are in Figure 2 and the stopping condition is $\varepsilon = 10^{-2}$ or the maximum 400 iterations. The decomposition results are compared with them using [4,5,8,11,12,15,18,26,39,40], respectively.

### 4.2. Image Decomposition Results Discussion

Figure 3 displays the decomposition results for Synthetic image. The parameters for the proposed model are $\mu = 0.1$, $\beta = 0.04$, $\tau = 1$, $\lambda = 0.2$, $\theta = 0.5$. The parameters in [8] are taken $\tau = 0.002$, $\mu = 0.001$, $\beta_1 = 1$, $\beta_2 = 0.1$, $\beta_3 = 0.001$. The parameters in [15] are $\gamma = 1$, $\alpha = 0.1$, $\mu_1 = 1$, $\mu_2 = 0.1$, $\mu_3 = 0.01$. From Figure 3, we can see that the new model gains much better decomposition results and some edges are kept in the cartoon image. The other methods contain cartoon contours (edges) which were wrongly assigned to the texture component. For the pseudo edge function, we only give the results of [40] and ours, since other methods without it. From Figure 3d,e, we can see that the pseudo edge image

appears many false edges in Figure 3d which are not consistent with the decomposed cartoon image (see some circular areas in Figure 3d), however the proposed method obtains the better results. In short, the proposed method is more effective than other four methods.

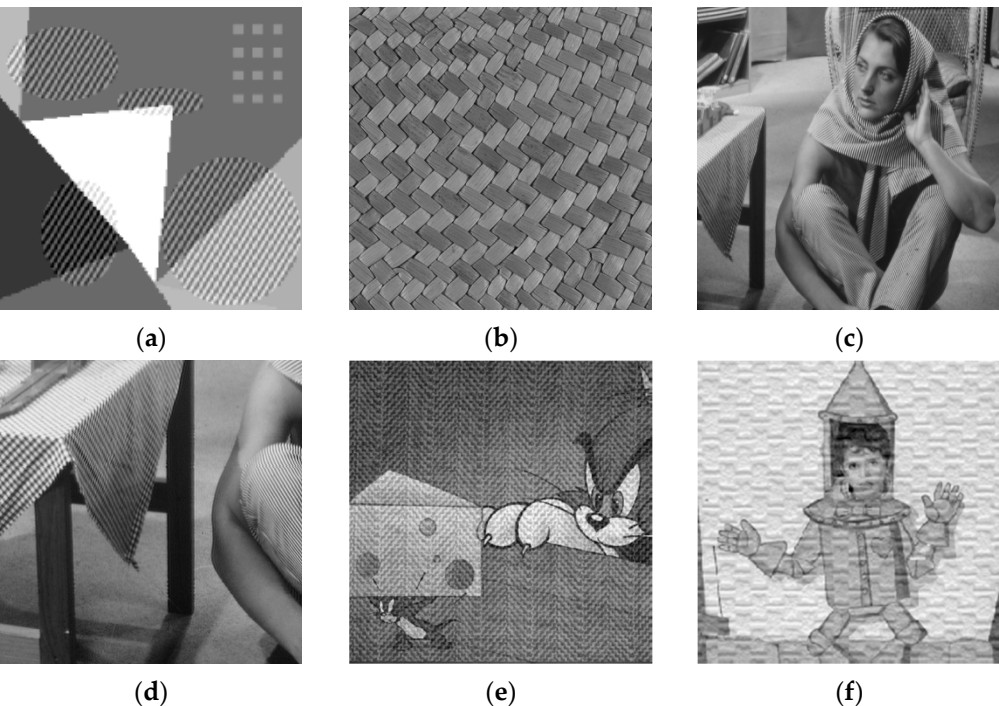

**Figure 2.** The test images. (**a**) Synthesis image; (**b**) Fabric image; (**c**) Barbara image; (**d**) Table image; (**e**) Tom image; (**f**) Boy image.

Figure 4 takes Fabric as the tested image, and the experimental parameters are taken $\mu = 0.04$, $\beta = 0.01$, $\tau = 1$, $\lambda = 0.4$, $\theta = 0.5$, respectively. Those parameters in [8] are manual mediated to achieve good results, which are selected $\tau = 0.001$, $\mu = 0.001$, $\beta_1 = 1$, $\beta_2 = 0.001$, $\beta_3 = 0.01$. The parameters in [39] are $K = 5$, $\lambda = 0.2$, $\Delta t = 0.1$. For the parameters in [40], $\lambda = 0.2$, $\mu = 0.1$, $\theta = 5$, $\Delta t = 0.02$ are chosen. From Figure 4a–d, we can see that the texture images in Figure 4b,c contain more contours which should be retained in the cartoon image. However, these characteristics are not obvious in Figure 4a,b. In addition, from the perspective of vision, the separated cartoon by the proposed method is better than [8]. For the pseudo edge image in Figure 4, we can observe that the obtained pseudo edge using [40] doesn't match the cartoon image.

In Figure 5, we take Barbara image to test the different methods. The experimental parameters are $\mu = 0.05$, $\beta = 0.04$, $\tau = 2$, $\lambda = 0.4$, $\theta = 0.5$. The parameters in [15,40] are $\gamma = 2$, $\alpha = 0.1$, $\mu_1 = 0.1$, $\mu_2 = 0.01$, $\mu_3 = 1$ and $\lambda = 0.2$, $\mu = 0.2$, $\theta = 2$, $\Delta t = 0.1$, respectively. From the decomposed cartoon images, we can observe that the cartoon image in Figure 5a contains some textures which are not extracted in the upper right area; the cartoon image in Figure 5d is smoother than other cartoon images. For the decomposed texture images, we can observe that the texture in Figure 5b is enhanced, and the texture in Figure 5d contains more contour structure which should be in the cartoon. Relatively speaking, the proposed method achieves better results. For the pseudo edge images in Figure 5d,e, we can see that the pseudo edge image in Figure 5d is not consistent with the cartoon image in Figure 5d, which can be seen from the neckline area of the Barbara image, while the proposed method better consistent with its cartoon image which can be seen from the Figure 5e.

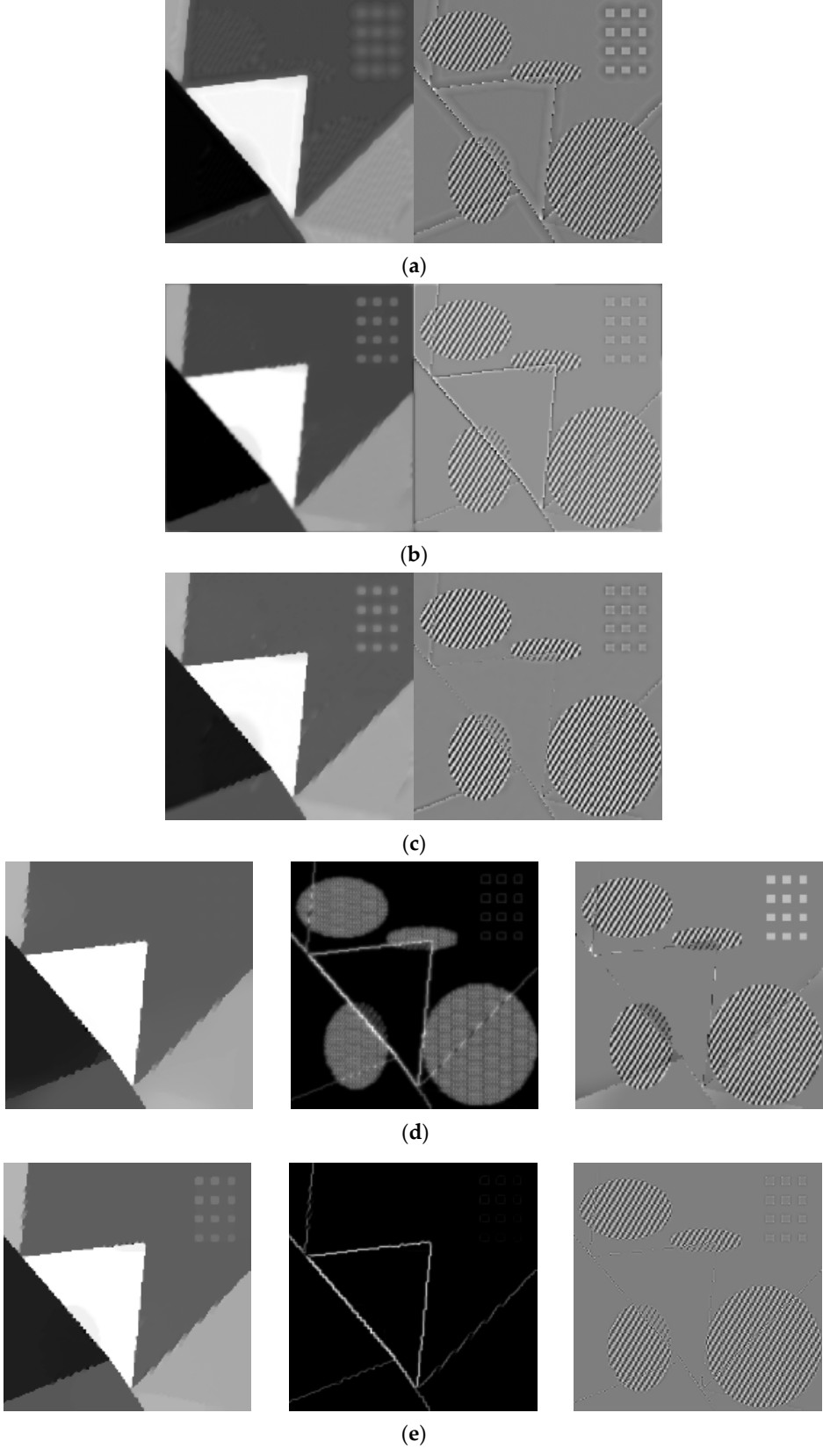

**Figure 3.** The decomposition results using different algorithms. From left to right: (**a**) Cartoon and texture obtained by [4]. (**b**) Cartoon and texture obtained by [8]. (**c**) Cartoon and texture obtained by [18]. (**d**) Cartoon, the pseudo edge and texture obtained by [40]. (**e**) Cartoon, the pseudo edge $|\omega|$ and texture obtained by the proposed method.

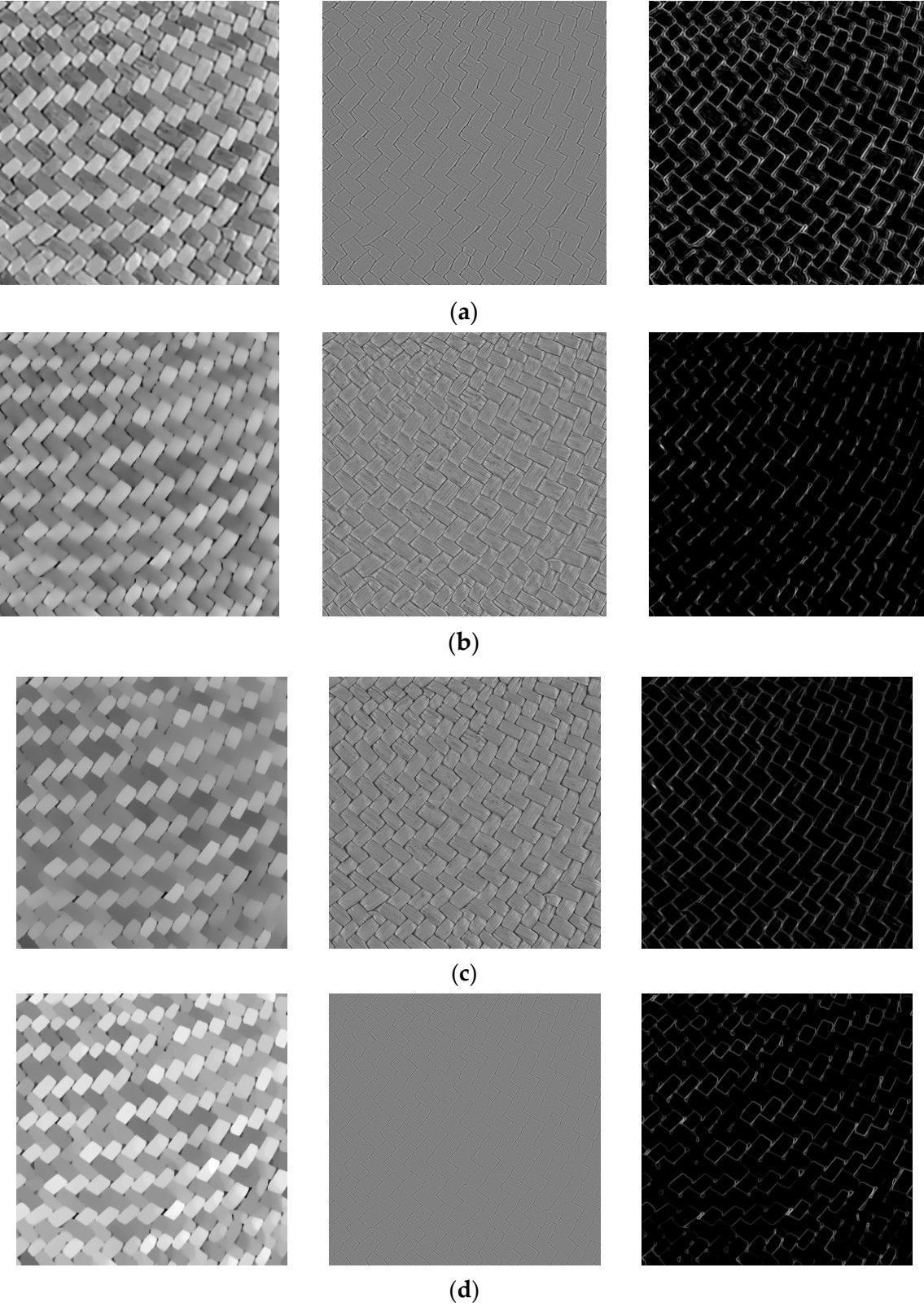

**Figure 4.** The decomposition results using different algorithms. From left to right: (**a**) Cartoon, texture and the module of gradient for the cartoon obtained by [8]. (**b**) Cartoon, texture and the pseudo edge obtained by [39]. (**c**) Cartoon, texture and the pseudo edge obtained by [40]. (**d**) Cartoon, texture and the pseudo edge $|\omega|$ obtained by the proposed method.

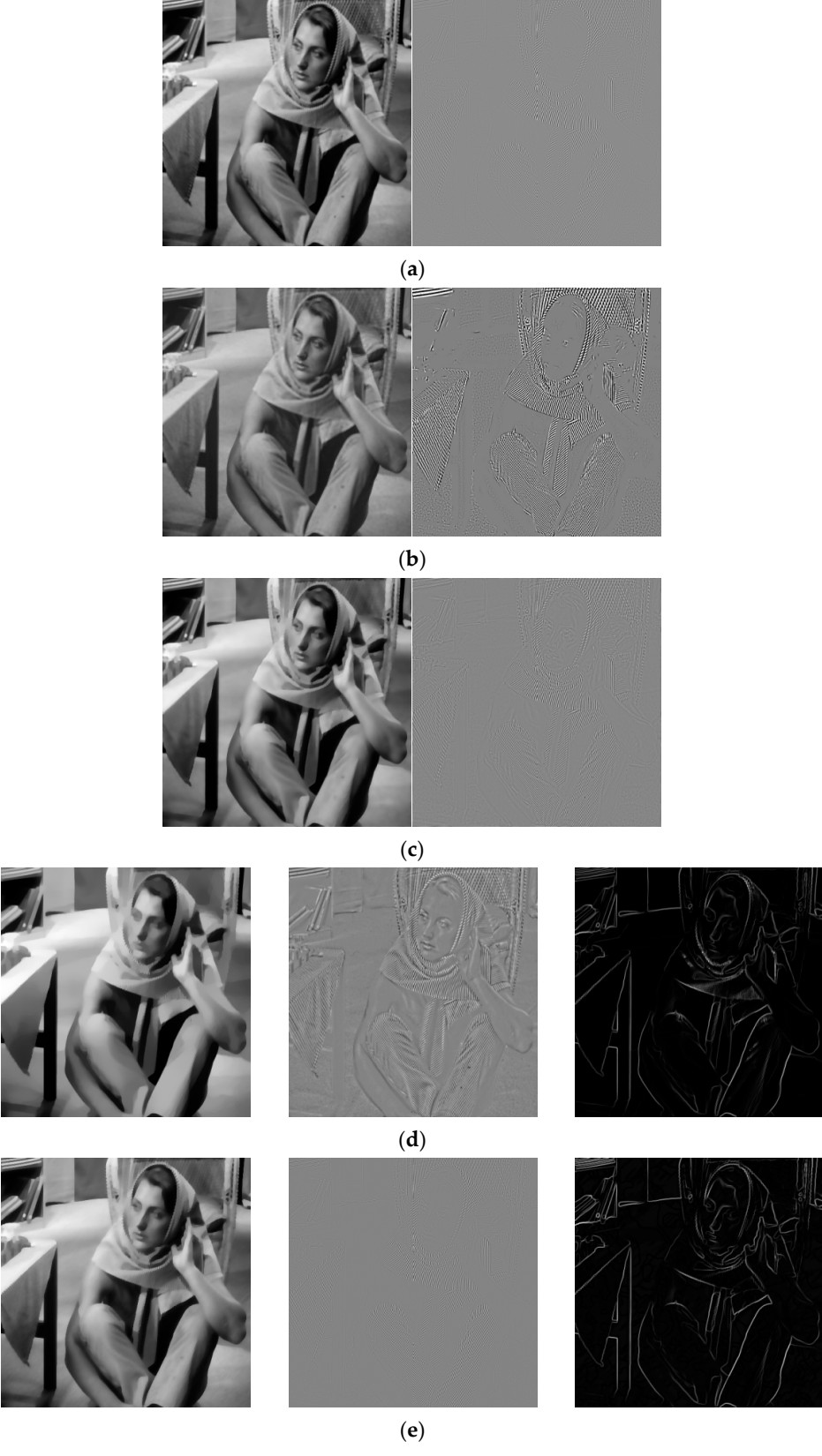

**Figure 5.** The decomposition results using different algorithms. From left to right: (**a**) Cartoon and texture [5]. (**b**) Cartoon and texture obtained by [26]. (**c**) Cartoon and texture obtained by [15]. (**d**) Cartoon, texture and the pseudo edge obtained by [40]. (**e**) Cartoon, texture and the pseudo edge $|\omega|$ obtained by the proposed method.

Figure 6 displays the decomposition results for Table image, which has a high presence of texture combined with cartoon parts. The parameters in [8,12] are chosen $\beta = 0.5$, $\mu_1 = \mu_2 = 0.1$, $\mu_3 = 0.01$, $\gamma = 10$ and $\tau = 0.002$, $\mu = 0.001$, $\beta_1 = 1$, $\beta_2 = 0.1$, $\beta_3 = 0.001$, respectively. The decomposition results in [26] can be run from the author's homepage, the scale is chosen 3 to reach the best result in their homepage. The parameters are $\tau = 2$, $\lambda = 0.1$, $\theta = 0.5$, $\mu = 0.05$, $\beta = 0.02$ for the proposed method. Visually, the cartoon images in Figure 6a,b are more blurry; [26,40] smooth the non-textured part, such as the legs and the arm; the texture image in Figure 6c is relative good; for the pseudo edge images in Figure 6d,e, the new method obtain the better results which can be observed from the red rectangular area. In a word, the new model performs comparable results with the other methods.

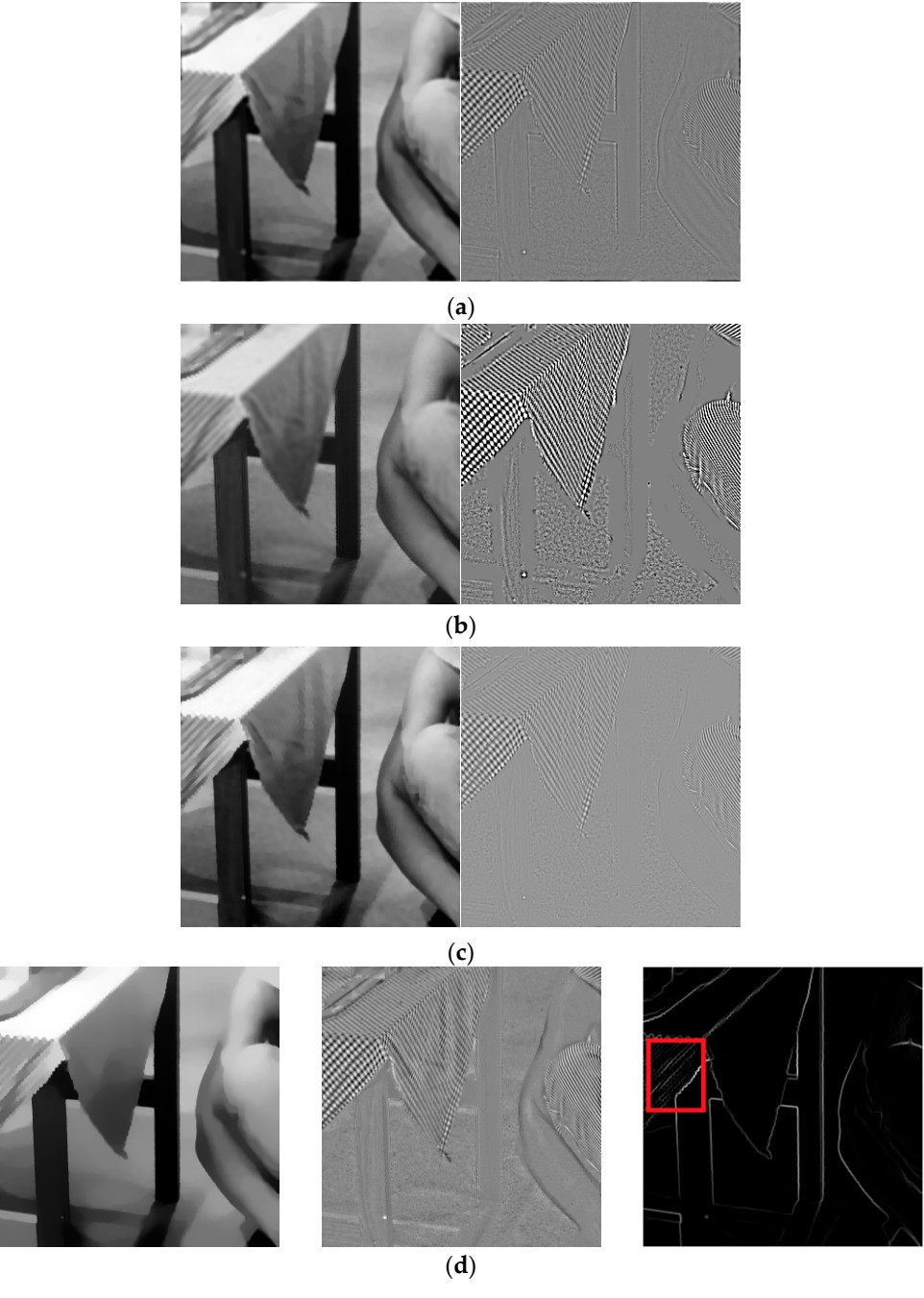

(a)

(b)

(c)

(d)

**Figure 6.** *Cont.*

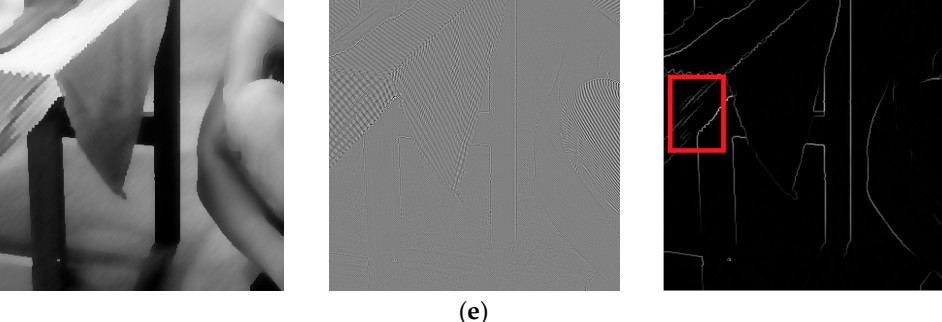

(**e**)

**Figure 6.** Table decomposition experiment. From left to right: (**a**) cartoon, texture with [8]; (**b**) cartoon, texture with [26]; (**c**) cartoon, texture with [12]; (**d**) cartoon, texture and the pseudo edge with [40]; (**e**) cartoon, texture and the pseudo edge $|\omega|$ obtained by the proposed method.

Figures 7 and 8 are the experimental results for Tom and Boy images, respectively. The parameters are $\tau = 1$, $\lambda = 0.2$, $\theta = 0.5$, $\mu = 0.1$, $\beta = 0.01$ and $\tau = 1$, $\lambda = 0.2$, $\theta = 0.5$, $\mu = 0.1$, $\beta = 0.04$ for the proposed method, respectively. The parameters are chosen to get the better results in [11]. The parameters are $\lambda = 0.2$, $K = 5$, $\Delta t = 0.1$ and $\lambda = 0.2$, $K = 3$, $\Delta t = 0.1$ in [39], respectively. The parameters are in [40] $\lambda = 0.2$, $\mu = 0.2$, $\theta = 5$, $\Delta t = 0.1$ and $\lambda = 0.2$, $\mu = 0.2$, $\theta = 1$, $\Delta t = 0.1$, respectively. Because [11,39] mainly studied image restoration, the residue $f - u$ denote the texture. Form Figures 7 and 8, we can see that the cartoon images and the pseudo edges by three methods are similar visually and the texture images obtained by [11,39,40] contain obvious cartoon information, i.e., cartoon and texture are not completely separated. In comparison, the new method achieves better results.

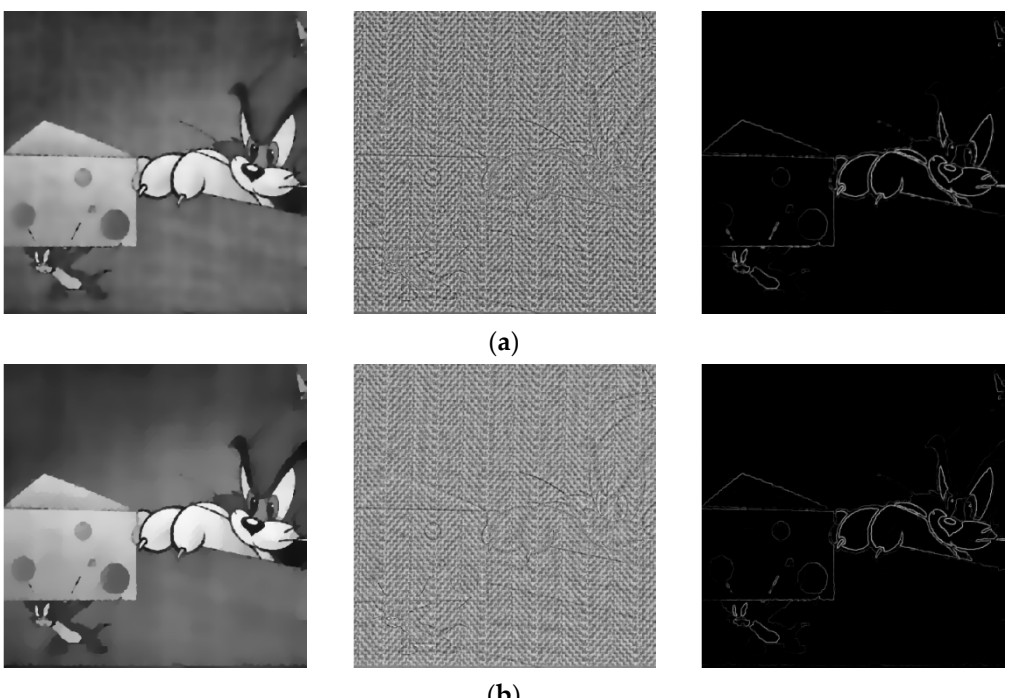

(**a**)

(**b**)

**Figure 7.** *Cont.*

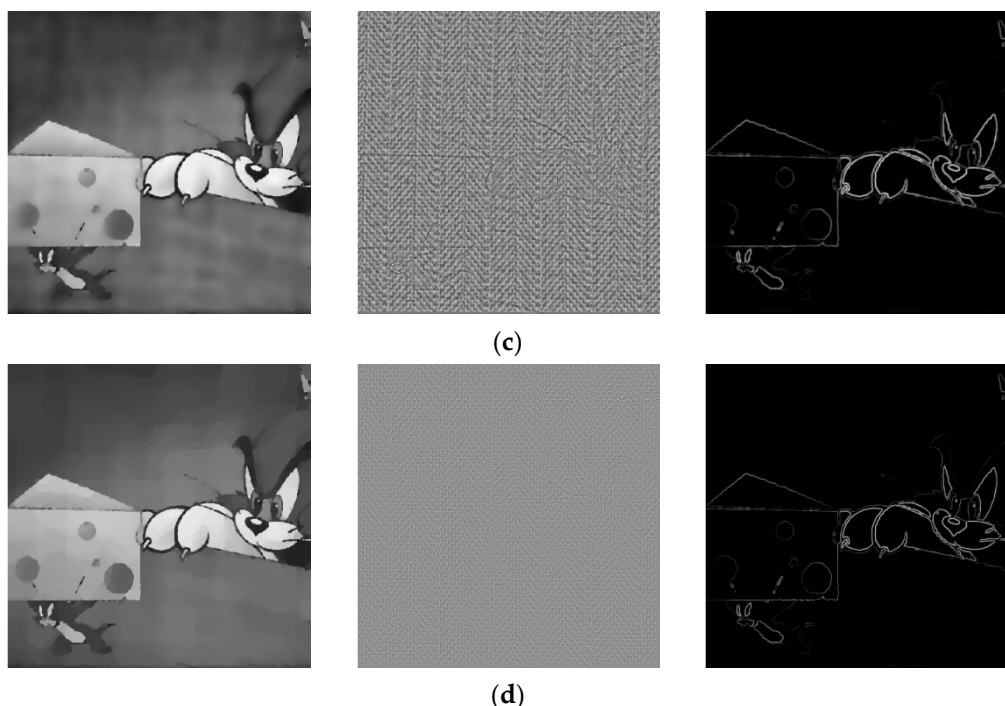

**Figure 7.** Tom decomposition experiment. From left to right: (**a**) cartoon, texture and the pseudo edge with [39]; (**b**) cartoon, texture and the pseudo edge with [40]; (**c**) cartoon, texture and the pseudo edge with [11]; (**d**) cartoon, texture and the pseudo edge $|\omega|$ obtained by the proposed method.

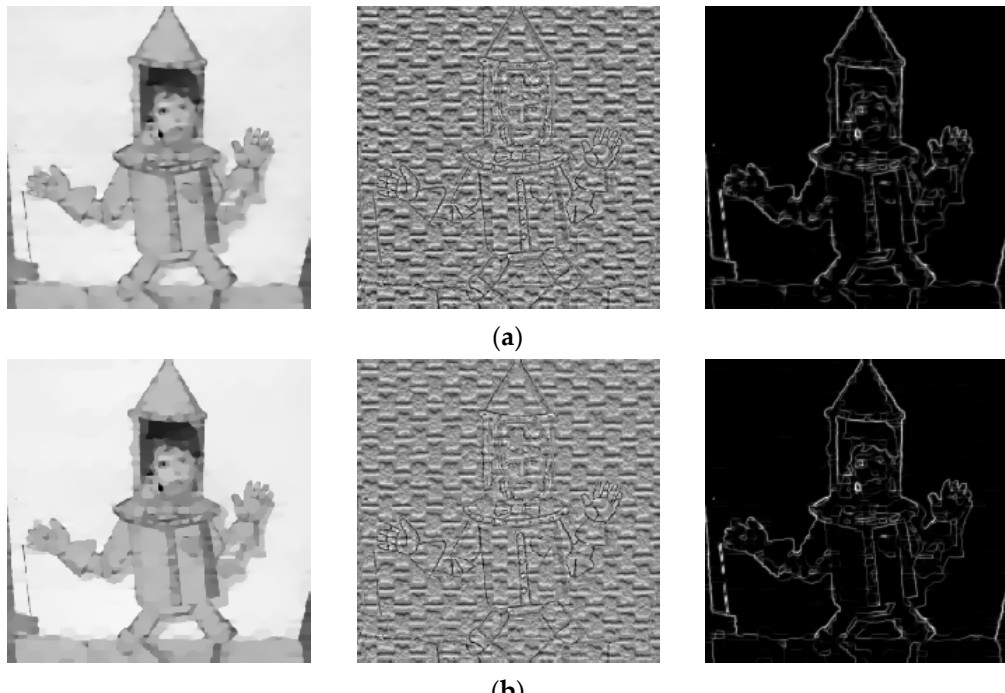

**Figure 8.** *Cont.*

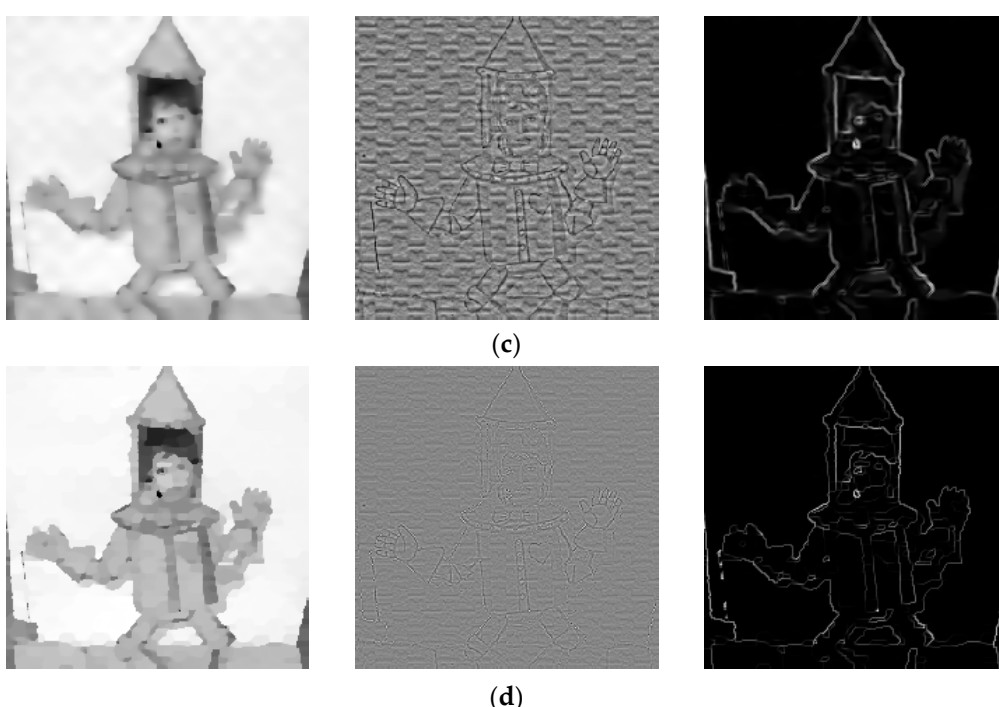

**Figure 8.** Boy decomposition experiment. From left to right: (**a**) cartoon, texture and the pseudo edge with [39]; (**b**) cartoon, texture and the pseudo edge with [40]; (**c**) cartoon, texture and the pseudo edge with [11]; (**d**) cartoon, texture and the pseudo edge $|\omega|$ obtained by the proposed method.

## 5. Conclusions

In this paper, one coupled variational system for image decomposition is proposed. In this coupled system, the vector field, structure and texture images are interwove. We design the alternating direction method, primal-dual method and Gauss-Seidel iteration for the given system. Using the proposed method, we can not only separate out the cartoon and texture parts, but the pseudo edges are extracted. Numerical experiments show the effectiveness of the proposed method compared with other variational methods.

**Author Contributions:** Conceptualization, J.X., Y.G., Y.H. and L.H.; methodology, J.X. and Y.H.; software, J.X., Y.G. and Y.H.; validation, J.X., Y.G. and L.H.; formal analysis, J.X. and Y.G.; investigation, J.X., Y.G., Y.H. and L.H.; resources, J.X., Y.G. and L.H.; data curation, J.X. and Y.G.; writing—original draft preparation, J.X. and Y.H.; writing—review and editing, J.X., Y.G. and L.H.; visualization, J.X. and Y.H.; project administration, J.X.; funding acquisition, J.X. and L.H. All authors have read and agreed to the published version of the manuscript.

**Funding:** This research was funded by the National Natural Science Foundation of China (No. U1504603), Key Scientific Research Project of Colleges and Universities in Henan Province (No. 22A120006), Guangxi Natural Science Foundation under Grant (No. 2018GXNSFBA281086).

**Institutional Review Board Statement:** Not applicable.

**Informed Consent Statement:** Not applicable.

**Data Availability Statement:** The data and the code are available on demand.

**Acknowledgments:** We would like to thank all the reviewers for their helpful suggestions.

**Conflicts of Interest:** All authors declare no conflict of interest.

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
