# Peer review of "A Coupled Variational System for Image Decomposition along with Edges Detection"

_algorithms, doi:10.3390/a15080288_

Round 1

Reviewer 1 Report

The manuscript deals with image restoration techniques, with special attention to the detenction of edges and to the separation of the image from the texture.

The manuscript should be checked substantially because the language is sometimes difficult to understand. As an example I rewrite the abstract

Abstract: In order to better decompose the images and protect their edges, in this paper we propose a coupled variational system consisting of two steps. The first step, an improved weighted variational model, is employed to obtain the cartoon and texture. Using the obtained cartoon image, in the second step, a new vector function is obtained for describing the pseudo edge of the considered image by one Tikhonov regularization variational model. Because a Tikhonov regularization model is equivalent to carrying out a Gaussian linear filtering, the obtained vector function can overcome the influence of outliers. To solve the coupled system, we give the alternating direction method, primal-dual method and Gauss-Seidel iteration. Using the coupled system, we can not only separate out the cartoon and texture parts, but also extract the edge. Extensive numerical experiments are provided in order to show the effectiveness of the proposed method compared with other variational methods.

Regarding still the Abstract, what are the outliers? In imaging and in mathematics more generally this word is used with quite different meanings and hence the authors have to specify.

Regarding the techniques described in the manuscript there is no mention on the boundary conditions which is a topic when dealing signals and images (see [HansenNagyOL,SSC] for the most used and successuful). Which are the boundary conditions here? In general the choice of not optimal boundary conditions could lead to ringing effects which spoil the quality of the image reconstructed by the used algorithms. Please discuss this point with regards to the references mentioned above and reported below.

[HansenNagyOL] Hansen, Per Christian; Nagy, James G.; O'Leary, Dianne P. Deblurring images. Matrices, spectra, and filtering. Fundamentals of Algorithms, 3. Society for Industrial and Applied Mathematics (SIAM), Philadelphia, PA, 2006.

[SSC] Serra-Capizzano, S. A note on antireflective boundary conditions and fast deblurring models. SIAM J. Sci. Comput. 25 (2003/04), no. 4, 1307–1325.

Author Response

Thank you very much for your suggestions. See the attachment for specific answers。

Reviewer 2 Report

The contribution presents a coupled variational system that contains of two steps: (1.) an improved weighted variational model is used to obtain the cartoon and texture. (2.) using the obtained cartoon image, a new vector function for describing the pseudo edge of an image by one Tikhonov regularization variational model is acquired. The paper needs to be proof read by a native speaker.

Author Response

(The authors gave the same response as above.)

Round 2

Reviewer 1 Report

the manuscript can be published